# Diffusivity Measurement by Single-Molecule Recycling in a Capillary Microchannel

**DOI:** 10.3390/mi12070800

**Published:** 2021-07-06

**Authors:** Bo Wang, Lloyd M. Davis

**Affiliations:** 1Center for Laser Application, University of Tennessee Space Institute, 411 B H Goethert Pkwy, Tullahoma, TN 37388, USA; 2Department of Physics and Astronomy, University of Tennessee Knoxville, 1408 Circle Dr, Knoxville, TN 37996, USA; ldavis@utsi.edu

**Keywords:** capillary, microfluidic device, single-molecule recycling, maximum likelihood

## Abstract

Microfluidic devices have been extensively investigated in recent years in fields including ligand-binding analysis, chromatographic separation, molecular dynamics, and DNA sequencing. To prolong the observation of a single molecule in aqueous buffer, the solution in a sub-micron scale channel is driven by a electric field and reversed after a fixed delay following each passage, so that the molecule passes back and forth through the laser focus and the time before irreversible photobleaching is extended. However, this practice requires complex chemical treatment to the inner surface of the channel to prevent unexpected sticking to the surface and the confined space renders features, such as a higher viscosity and lower dielectric constant, which slow the Brownian motion of the molecule compared to the bulk solution. Additionally, electron beam lithography used for the fabrication of the nanochannel substantially increases the cost, and the sub-micron dimensions make the molecule difficult to locate. In this paper, we propose a method of single-molecule recycling in a capillary microchannel. A commercial fused-silica capillary with an inner diameter of 2 microns is chopped into a 1-inch piece and is fixed onto a cover slip. Two o-rings on the sides used as reservoirs and an o-ring in the middle used as observation window are glued over the capillary. The inner surface of the capillary is chemically processed to reduce the non-specific sticking and to improve capillary effect. The device does not require high-precision fabrication and thus is less costly and easier to prepare than the nanochannel. 40 nm Fluospheres^®^ in 50% methanol are used as working solution. The capillary is translated by a piezo stage to recycle the molecule, which diffuses freely through the capillary, and a confocal microscope is used for fluorescence collection. The passing times of the molecule through the laser focus are calculated by a real-time control system based on an FPGA, and the commands of translation are given to the piezo stage through a feedback algorithm. The larger dimensions of the capillary overcomes the strong sticking, the reduced diffusivity, and the difficulty of localizing the molecule. We have achieved a maximum number of recycles of more than 200 and developed a maximum-likelihood estimation of the diffusivity of the molecule, which attains results of the same magnitude as the previous report. This technique simplifies the overall procedure of the single-molecule recycling and could be useful for the ligand-binding studies in high-throughput screening.

## 1. Introduction

Techniques such as fluorescent labeling have enabled the behavioral observation of a single molecule without the description of ensemble thermodynamics, so that enhanced the precision of localizing single beads and conquered the diffraction limit of a conventional microscope [1,2]. To improve the signal-to-noise ratio of the single-molecule spectroscopy, a spatial pinhole is used to block out-of-focus light and adapt the focal volume to the size of a point detector, single-photon avalanche diode (SPAD), which measures the time of each photon down to sub-nanosecond precision [3]. However, the Brownian motion limits the time the molecule stays inside the focal volume and thus the observation time is restricted to the scale of milliseconds, which is not enough to witness changes, such as protein folding and molecular interactions. To overcome this disadvantage, surface tethering and optical trapping are developed to use chemical and electromagnetic attractions to immobilize the molecule but alter the behavior in the meantime [4,5]. Feedback-driven tracking and trapping methods measure the displacement of the molecule from the center of the focal volume and apply real-time compensation to the position of the sample or laser focus [6,7,8,9,10,11,12,13]. However, the feedback control requires the molecule to fluorescent continuously and thus the observation during is limited by photobleaching and photoblinking [9].

An alternative method to prolong the observation is to translate a single molecule through the laser focus with a constant velocity and repeat the translation after a fixed delay [14], so that the molecule is periodically and most of the time in the dark and has opportunity for recovery from the reversible dark states, and thus the time before irreversible photobleaching is highly extended [14]. This technique, namely, single-molecule recycling (SMR), can extend the overall observation duration to more than 10s [14]. However, there are rigorous requirements for the inner surface of the nanochannel to reduce the tethering. Features such as hydrophilicity have to be adjusted to reduce the interaction between the surface and the analyte molecule and to optimize the electrophoresis for the recycling [14,15,16]. Another disadvantage of the tube with nanoscale inner diameter is that the diffusivity of the molecule is lowered by the more frequent interaction to the surface, which renders a higher viscosity and lower dielectric constant [15,17], compromising the studies of single molecules in aqueous environment. Additionally, the sub-micron dimensions of the nanochannel make the molecule difficult to locate.

Fused-silica capillary is an alternative to nanochannel created by electron-beam lithography and has been widely used in pharmaceutical industry and academic research. In the previous studies, a single molecule is detected in a sub-micron capillary at room temperature, so that the Brownian motion of the molecule is slowed down by the restriction of the channel and observation 50–100 times longer than the bulk diffusion time is achieved [15]; the capture antibody is immobilized on the inner surface of the capillary to study its interaction with detector antibody, and the procedure is carried out by a lab-on-a-chip device containing capillaric circuits, which are also applicable to DNA analysis and DNA amplification [18]. In this paper, we use a microchannel made from fused silica capillary with an inner diameter of 2μm and an outer diameter of 150μm to recycle the molecule. Because the inner diameter of the capillary is significantly larger than that of a nanochannel, the interaction between the molecule and the surface is substantially reduced, and thus the sticking is largely eliminated. Consequently, the molecule moves with greater diffusivity, which simplifies the monitoring of the size and the shape of the molecule. However, the length of the channel is in the scale of millimeters, so that higher voltage and insulated sample plate are required by driving the solution with electrophoresis. Alternatively, we use a piezo stage to translate the device and use the locations of the molecule as the centers of recycling. A calibration based on fluorescence correlation spectroscopy (FCS) is employed to calculate the translation speed of the piezo stage [19]. A digital filter [20] is developed to discriminate molecular photon burst from background and the passing time of the molecule is calculated by peaks of the filtered signal. A LabVIEW Real-Time system including a Field-Programmable Gate Array (FPGA) implements the peak detection algorithm and translates the piezo stage to recycle the molecule after a fixed delay [16] even when the anticipated burst is not detected. The control system attains a time precision of 10μs, which is much faster than the 2ms time resolution reported by Lesione et al. [14]. A maximum number of recycles of more than 200 is realized by this setup, which gives an observation duration of around 6 s.

The diffusivity of a molecule depends on its size and interactions with the suspending system. In high-throughput screening and pharmaceutical drug discovery research, a speed-up in the diffusivity of a fluorescent ligand when it becomes unbound from a target biomolecule indicates the competitive binding of one of the molecules from the library being screened [21]. The feedback-driven tracking and SMR are applicable to measure the diffusion coefficient of a single molecule. The feedback-driven tracking measures the diffusivity by fitting the mean-square displacement of the molecule’s trajectory, and the precision of the estimation depends on the during of the trajectory [22,23,24,25]. SMR estimates the diffusivity from fluctuations in the intervals between detected passages, and the precision depends on the number of times the molecule is recycled [26]. In this paper, we have developed an maximum-likelihood (ML) method to estimate the diffusivity of a single molecule from the times of the detected passages and the times the piezo stage starts to move, which is based on the estimation used by SMR in a nanochannel [26], and attained results in the same magnitudes as previous reports [9]. The work in [26] makes the estimation using variations of time intervals between molecule passings; this paper uses displacements of the molecule, which is described by Equation (Equation 10), to estimate the diffusivity. The technique covered by this paper simplifies the overall procedure of SMR and could be useful to monitor the change of diffusion coefficient for the ligand-binding studies.

## 2. Materials and Methods

### 2.1. Molecule Diffusing in a Microfluidic Channel

When a molecule is confined to a microfluidic channel, it is free to move in the axial dimension. The Brownian motion of the molecule and transport of flow can be described by a one dimensional Fokker–Planck equation [14], which is given by
(1)$$\frac{\partial p(x,t)}{\partial t}=D\frac{\partial^{2}p\left(x,t\right)}{\partial x^{2}}-v\frac{\partial p(x,t)}{\partial x},$$
where p(x,t)dx denotes the probability density function (pdf) to find the molecule within dx of *x* at time *t*, *D* denotes the Einstein–Stokes diffusion coefficient, and *v* denotes the flow velocity. The initial condition of the equation is p(x,t=0)dx=δ(x)dx, which depicts a molecule passes to the right through the center x=0 of the focus laser beam at time t=0. δ(x) is Dirac delta function, which is infinity when x=0. The shape of p(x,t) evolves to a Gaussian function with mean displacement due to flow and width increasing due to diffusion, which is
(2)$$p\left(x,t\right)dx=\frac{1}{\sqrt{2\pi }\sigma \left(t\right)}\exp\left[\frac{-1}{2}{\left(\frac{x-\mu (t)}{\sigma (t)}\right)}^{2}\right]dx,x\in \left(0,\infty \right),$$
where μ(t)=vt, $\sigma \left(t\right)=\sqrt{2Dt}$. For a freely diffusing molecule, v=0 and μ(t)=0, and if the time of diffusion is T, the width of the Gaussian function becomes $\sigma \left(T\right)=\sqrt{2DT}$.

### 2.2. Single Molecule Recycling to Measure Diffusivity

For SMR with a piezo stage, the position of the molecule is used as the center of the recycling, which is depicted as a burst of photon counts. The molecule in the capillary is moved back and force using the piezo stage, which is set a fixed distance from the center after a photon burst is detected and waits for a fixed delay *T* reduced by time of transition before reversing back to find the next photon burst. The mechanism of SMR is described by the inset of Figure 1 and will be discussed in Section 2.6. If the first molecular detection is the starting point of the pdf and there is no missed detection, the width of the pdf will be broadened to $\sqrt{2DT}$ after the second detection, which gives
(3)$$p\left(\Delta x,T\right)dx=\frac{1}{\sqrt{(}4\pi DT)}\exp\left(-\frac{\Delta x^{2}}{4DT}\right)dx,x\in \left(0,\infty \right),$$
where Δx is the distance the molecule diffuses. In principle, the diffusivity may be estimated from just a single measurement of Δx. To simplify the discussion, consider the case where the method used for estimating the displacement of the molecule between two passages is exact and where there is no missed detection. According to the ML method [27], we get a function that expresses the likelihood of the parameter *D* for the given measurement:(4)$$L\left(D;\Delta x\right)=\frac{1}{\sqrt{4\pi DT}}\exp\left(-\frac{\Delta x^{2}}{4DT}\right),D\in \left[0,\infty \right).$$

The ML estimate $\hat{D}$ is the value of *D* for which Equation (Equation 4) is a maximum, which is found by solving ∂L(D;Δx)/∂D=0, which gives the ML estimate
(5)$$\hat{D}=\frac{\Delta x^{2}}{2T}$$

### 2.3. Confocal Microscope System

In this paper, the optics system is a custom-built confocal fluorescence microscope through a beam expander, which is shown in Figure 1.

The laser used in the experiment is a mode-locked dye laser (Coherent 700) pumped by a 532 nm solid-state laser (Spectra Physics Vanguard). The dye laser uses DCM-special dye and produces picosecond pulses at 76 MHz at a wavelength of 647 nm. To adjust the collimation of the dye laser beam and expand it to give the desired beam size at the entrance pupil of the objective, it passes through a beam expander consisting of three lenses (Newport KPC031, KBX064, and KPC019), which produces a collimated beam with size (beam waist) 0.17 mm. The microscope uses a water immersion objective with NA 1.2 (Olympus UPLSAP060XW), a 250 mm focal length plano-convex lens as a tube lens for the microscope, and a long-pass dichroic filter (Omega 3RD670LP) to isolate the fluorescence from the scattered laser light and Raman-scattered light. After a sample is placed on the observation stage, the laser beam is focused into the sample by the objective so that molecules in the focal volume will emit fluorescence, then the fluorescence from the sample is collected by the same objective. A beam sampler splits the fluorescence to an EM-CCD camera (Andor iXon Ultra 897, Oxford Instruments) and the single-photon avalanche diodes (SPADs; custom units with detector heads from Perkin-Elmer, Canada, and circuits made by MPD, Italy). A 200-micron pinhole (Thorlabs P200H) is used to filter the high-frequency noise spherically distributed outside the center of the beam and thus to improve the signal-to-noise ratio (SNR) of the fluorescence signal. The EM-CCD can visualize fluorescently labeled beads when they pass through the excitation volume of the laser. In setting up the experiment, we first use a Kohler lens to defocus the laser wide-field illumination so we can view the microfluidic device on the EM-CCD camera and adjust its transverse position to position the center of the microchannel where the laser will focus. We then filp the Kohler lens out of the beam and focus the beam into the channel. As the EM-CCD and the pinhole for the SPADs are parfocal (i.e., they are the same distance from the plane of focus), a clear image of the passing beams indicates that the channel is in the confocal volume and that the SPADs should detect the fluorescence with high SNR. Fine adjustments to the piezo positioning are then made while monitoring the count rates of the SPADs.

### 2.4. Real-Time Control System

To accomplish SMR, we built a control system using LabVIEW Real-Time software. The control system determines the timings of photons detected by the SPAD and sends digital signals to adjust the position of the piezo nano-translation unit. The control system includes a NI PCI-7833R FPGA multifunction data acquisition card with digital inputs for signals from the SPADs and a NI PCI-DIO-96 digital input/output card that sends digital signals to the parallel input output (PIO) interface of the piezo stage controller. These two cards are contained with a target computer (Remote Desktop), which is connected to a host PC via ethernet. Figure 2 shows a description of the system.

Photons are detected by the SPAD, which generate transistor-transistor logic (TTL) pulses and their times are determined by the FPGA. We set up a 100 MHz clock on the FPGA and record the count of clock ticks whenever the SPAD sends a TTL pulse to the FPGA, thereby recording the timing of the photons with a precision of 10 ns. These data are used only for postprocessing, such as for calculation of the autocorrelation function, which will be discussed in Section 3. We also set up the FPGA to count the TTL pulses from the SPAD and to send the count to the target PC every 10μs. The target PC runs a LabVIEW program that calculates the weighted sliding sum (WSS; a matched filter that will be discussed in Section 3) from the counts received from the FPGA and finds the peaks of the WSS. The times to reverse the flow are determined by adding the reversal delay to the times of the WSS peaks and are sent back to the NI PCI-DIO-96 card to send out digital signals so that the sample is moved back and forth using the piezo stage. The commands for moving the piezo stage are within a timed loop that is synchronized to a 100 kHz clock signal generated by the piezo controller (PI E-710 console). This synchronization is implemented by counting piezo clock pulses at the FPGA and sending data from the FPGA to the target PC at 10μs intervals whenever the piezo clock count is incremented. In this way, the real-time program synchronizes its times with the possible motion of the piezo. There is a delay to initiate the piezo stage to move, which is less than 2 ms. Details of SMR using a piezo stage are covered in Section 2.6. The program on the host PC (which is called a virtual instrument or VI) sets up the experiment parameters and plots the count rate of photons, values of the peaks of the WSS, the position of the piezo stage, and histograms of the times between the start of the piezo stage’s motion and the peaks of the WSS. From the VI on the host PC, we can send commands to recenter the piezo stage or manually adjust its xyz position, which enables us to accurately position the microfluidic device with respect to the laser beam. During the experiment, we only change the x position of the piezo stage, which will be discussed in Section 2.6. We can also adjust the threshold for detecting peaks in the WSS and the delay time and other parameters for SMR while observing the experimental outcomes. The program on the target PC consists of the algorithm of SMR, the control of the piezo stage, and the data flows to the host PC.

### 2.5. Preparation of the Capillary Microchannel

The microchannel device is made from fused silica capillary (Molex TSP002150, ) with an inner diameter of 2μm and an outer diameter of 150μm, as seen in Figure 3a. The width of the laser beam is adjusted to the inner diameter of the tube to improve the SNR of the confocal detection, as shown in Figure 3b. The capillary has a thin polyimide coating, which gives a strong background fluorescence. To make the capillary microchannel, we first use a ceramic cleaving stone to chop the capillary into 1-inch pieces, then we remove the coating by baking for an hour at $700{\;}^{\circ }$C. We then use silicone glue to fix the stripped capillary onto a glass cover slip and also to fix three o-rings above the capillary to the cover slip, as seen in Figure 3c.

The two outer o-rings form reservoirs for the solution, and the o-ring in the middle is filled with the oil used for an immersion objective, which has the same refractive index as the cover slip (1.56). The oil has approximately the same refractive index as the fused silica capillary (1.46) and thus significantly reduces specular reflection and refraction at the cylindrical walls of the capillary. To improve the capillary effect, we need to improve the hydrophilicity of the inner surface. We first rinse the capillary with 0.1 M sodium hydroxide (NaOH) for 5 min, then rinse with distilled water for 2 min. To reduce the non-specific sticking of the molecule to the surface of fused silica, which is made worse by attraction between the charge on the surface and the polarity of the molecule, we process the capillary with Tween-20 detergent (Sigma-Aldrich, St. Louis, MO, USA) as follows [14]: We first make a 0.02% Tween-20 solution in 1 × TAE (40 mM Tris-acetate and 1 mM EDTA) aqueous solution, where the pH value is around 8.0. We rinse the capillary with it for 10 min until a layer of Tween-20 forms on the surface of fused silica. To make working solution, we use 50% methanol as solvent and 40 nm Fluospheres^®^ (Thermo Fisher) as the solute, which reaches a concentration of 10 nM and provides a high emission rate of fluorescence and low sticking to the surface. We pipette the working solution into the o-ring for solution and let the working solution substitute the buffer solution by capillary effect.

### 2.6. Experimental Practice

For SMR in the capillary microchannel, an algorithm is developed to move the capillary with the molecule within it back and force using the piezo stage, while adjusting the end points of the motion as the molecule diffuses along the capillary. To do so, the estimated position of the moelecule is used as the center of the recycling and the piezo stage starts to move from a position that is a prescheduled distance $X_{t}$ (10,000 piezo units in the experiment, which is equal to 4.58μm) away from the center. The piezo then remains at the destination for a certain period, which is equal to the reversal delay Δt minus the time of translation. Δt is set to different values and Δt=30ms gives the maximum number of recycles. After this, the piezo is moved in the opposite direction and we use the estimated position of the molecule reduced by $X_{t}$ as the destination. As the molecule diffuses in the microchannel, we must renew the estimated position of the molecule for each move. To estimate the current position of the molecule, we subtract the mean value of the distance between the time of the last WSS peak $t_{w}\left(i\right)$, where *i* is the index of the current control loop, and the starting time of the translation $t_{p}\left(i\right)$ from its current value and multiply this by a scale factor C, which has the unit of piezo units per millisecond and could be modified during the experiment to optimize the recycling. We then add the result to the previous center of the recycling to obtain a new estimated position for the molecule, as follows:(6)$$X\left(i+1\right)=\left[\left(t_{w}\left(i\right)-t_{p}\left(i\right)\right)-m\right]\times C+X\left(i\right),$$
where X(i) is the last estimated position of the molecule, X(i+1) is the estimated current position of the molecule, and m is the mean value of X(i+1)−X(i). The algorithm of SMR is described in Figure 4.

## 3. Results and Discussion

### 3.1. Measurement of the Translation Speed

To measure the diffusivity of molecules using SMR, we need to know the flow velocity *v*, which can be experimentally measured by fitting the normalized autocorrelation function (ACF) [21]. The ACF of fluorescent particles within a 3-dimensional Gaussian-shaped probed region was first reported by Elson et al. [28], and was adapted to constant flow speed by Magde et al. [19]. In our experiment, the fluorescence photons are collected by SPADs and processed by a correlator program to calculate the normalized ACF. The molecule is confined by the one-dimensional microchannel and passes through a Gaussian beam waist $\omega_{0}$ with constant velocity, and thus the fitting function of ACF is customized to one dimension with four fitting parameters $a_{0}$, $a_{1}$, $a_{2}$, $a_{3}$, which is given by
(7)$$g\left(\tau \right)=a_{0}+\frac{a_{1}}{\sqrt{1+a_{3}\tau }}\exp\left(\frac{-a_{2}\tau^{2}}{1+a_{3}\tau }\right),$$
where $a_{1}=F^{2}/{\left(F+B\right)}^{2}$, in which *F* is the rate of fluorescence counts from a molecule at the center of the laser focus and B is the background rate, and $a_{0}=1$, which gives g(∞)=1, $a_{2}={\left(v/\omega_{0}\right)}^{2}$ is the square of the reciprocal of the time constant for flow, and $a_{3}=4D/\omega_{0}^{2}$ is the reciprocal of the time constant for diffusion, in which *D* is the diffusivity of the molecule. We use the SMR control flow to measure the speed a molecule passes through the beam waist. We first immobilize 40 nm Fluospheres on a glass cover slip with PDMS and fix the cover slip on the piezo stage. After positioning a Fluoshphere into the center of the beam waist, we recycle the bead and calculate its ACF from the times of the detected photons. In this case, $a_{0}$ is the background level of the AF between peaks, which is equal to 0.75 from the fit of the ACF, $a_{1}$ is the amplitude of the peak of the ACF, $a_{3}=0$ as the bead does not diffuse and D=0, as seen in Figure 5.

The fit of the ACF gives $\omega_{0}/\left(2v\right)=0.47\;ms$. In this experiment, the laser beam is adjusted to fit the inner diameter of the capillary, which is 2μm, thus the beam waist $\omega_{0}$ is estimated to be 1μm. This gives a translation speed of $v=1.1\times 10^{-3}\;ms^{-1}$. We use $v=2.0\times 10^{-3}\;ms^{-1}$ as the beam size is usually larger than the inner diameter of the capillary due to defocusing of the objective.

### 3.2. Matched Filter for Photon Burst Detection

To detect the passage times of molecules through the laser focus, photon bursts must be recognized above the background. We have reported an algorithm based on matched filter for photon burst detection in the previous paper [26], where simulations are made to investigate the threshold of photon bursts and the width of the weight function. To detect photon bursts, we first process the stream of photon times by a weighted sliding sum (WSS), in which the weights are proportional to the expected temporal profile of the fluorescence signal as a molecule passes through the laser focus. Thus, the WSS ideally corresponds to a matched digital filter [20]. For a molecule passing at constant velocity *v* through a Gaussian laser focus of waist $\omega_{0}$, the weights are given as
(8)$$w\left(t\right)=A\exp\left[-\frac{{\left(t-3\sigma_{t}\right)}^{2}}{2\sigma_{t}^{2}}\right],$$
where the width of the weight function is ideally $\sigma_{t}=\omega_{0}/\left(2v\right)$, and A is the amplitude of the weight function, which is taken as 128.0 in the control program. The WSS can be computed in real time in LabVIEW Real-Time program with NI-PCI-7833R FPGA data acquisition card at discrete intervals of $\Delta t_{w}=10\;\mu s$, and thus the time resolution of detection is 10μs. After a WSS above the threshold is detected, the following WSS values are stored in an array until a WSS is below the threshold. Then, the time corresponds to the maximum value in the WSS array is taken as the occurrence time of the photon burst. Figure 6 illustrates the photon bursts of a Fluosphere processed by WSS. The Fluosphere diffuses through the capillary and is translated by the piezo stage, the intervals between peaks center around Δt and are differed from error induced by diffusion.

### 3.3. Diffusivity Measurement from Single-Molecule Recycling in a Capillary Microchannel

As seen in Figure 4, the algorithm uses the position of the molecule as the center of recycling, thus the position of the molecule in respect to the capillary can be estimated. During the experiment, $t_{w}\left(i\right)$ from Equation (Equation 4) is measured by the FPGA clock and is stored in one data file. X(i) is calculated iteratively and is stored in the same file. From the collected data, the position of the molecule during the course of each translation can be derived using the following equation:(9)$$X_{m}\left(i\right)=X\left(i\right)-{\left(-1\right)}^{i}\left[\left(t_{w}\left(i\right)-t_{p}\left(i\right)\right)v-X_{t}\right],$$
where $X_{m}\left(i\right)$ is the position of the molecule, and *v* is the translation speed. Figure 7a depicts the trajectory of the piezo stage and the trajectory of the molecule, in which the blue line describes the piezo stage stops at one end of the path and waits until the next command of recycling and then translates the molecule through the laser focus, the red dotted line shows the molecule diffuses during the reversal delay and the centers of recycling are displaced in respect to the locations of the molecule. Figure 7b shows the difference between the WSS peak $t_{w}\left(i\right)$ and the starting time of the translation $t_{p}\left(i\right)$ of the same period as Figure 7a, which represents the contribution of diffusion in estimation the position of the molecule according to Equation (Equation 9). Because the capillary is translated to one direction to detect the molecule, a too large $t_{w}\left(i\right)-t_{p}\left(i\right)$ will exceed the limit of motion of the piezo stage and jeopardize the algorithm and hence discontinue the recycling.

Substituting $\Delta X\left(i\right)=\left(t_{w}\left(i\right)-t_{p}\left(i\right)\right)v$ into Equation (Equation 9), we find the variation of position caused by diffusion is given by
(10)$$X_{m}\left(i+1\right)-X_{m}\left(i\right)=\left(X(i+1)-X(i)\right)+{\left(-1\right)}^{i}\left(\Delta X\left(i+1\right)+\Delta X\left(i\right)-2X_{t}\right).$$

According to Equation (Equation 5), the diffusivity from a single measurement ${\hat{D}}_{i}$ is obtained from $\sqrt{2{\hat{D}}_{i}T}=X_{m}\left(i+1\right)-X_{m}\left(i\right)$, where *T* is the reversal delay. Substituting Equation (Equation 10) gives
(11)$${\hat{D}}_{i}=\frac{1}{2T}{\left[\left(X(i+1)-X(i)\right)+{\left(-1\right)}^{i}\left(\Delta X\left(i+1\right)+\Delta X\left(i\right)-2X_{t}\right)\right]}^{2}.$$

The theory of ML gives $\hat{D}=\sum_{i}{\hat{D}}_{i}/N$ [27], where *N* is the number of measurements. Therefore, the estimation of diffusivity becomes
(12)$$\hat{D}=\frac{1}{2NT}\sum_{i}{\left[\left(X(i+1)-X(i)\right)+{\left(-1\right)}^{i}\left(\Delta X\left(i+1\right)+\Delta X\left(i\right)-2X_{t}\right)\right]}^{2}.$$

To process the data files, we first reconstructed the WSS with the photon timings and recorded the peaks of bursts. Then, we made a binned histogram of the time between the peaks and the starts in the FPGA clock and fit with Gaussian function. As the piezo stage sometimes reaches its limit of motion and stops at the ends according to the algorithm, the molecule diffuses to the laser beam without the translation, which can cause the time between the peak and the start to be too short or too long. Therefore, we took the Gaussian function centered around 2.5 ms as the effective distribution and select the translations within a 3 sigma width of its peak. From Equation (Equation 12), we can estimate the diffusivities of individual molecules. The estimated diffusivities, which center at approximately $5\times 10^{-11}\;m^{2}s^{-1}$, reach the same magnitude as the value from previous report ( $1.2\times 10^{-11}\;m^{2}s^{-1}$) [9], where the diffusivity of 40 nm Fluosphere in water-based solution is precisely studied for the first time. The larger diffusivity in this paper could be caused by adding Tween-20 to the running buffer and error in calibrating the translation speed. As seen in Figure 8, variation of the centers of recycles and the estimated diffusivities of individual FLuospheres from 4 experiments are illustrated. The estimated diffusivities follow a Gaussian distribution and the maximum number of recycles is above 200, which gives a precision of about ±10% [26].

During the experiment, the algorithm may loss track of the molecule caused by photobleaching or diffusion, and then a new molecule will be loaded to the recycling by constant flow or prolonged translation of the piezo stage. Because the microchannel has larger dimensions than the nanochannel, a molecule has higher probability to enter the channel, and the less interaction to the surface enables the molecule to move faster through the channel and thus locating of the molecule takes less time. We use the 40 nm Fluosphere in 50% methanol solution in the nanochannel device to apply SMR [16] and compare the time to locate a new molecule for SMR in nanochannel and SMR in microchannel, which is shown in Figure 9. The median time to locate the molecule is 80.1 ms for the nanochannel and 4.8 ms for the microchannel. The time ranges from 80 ms to 1 s for the nanochannel and from 3.8 ms to 10 ms for the microchannel. Overall, the microchannel sufficiently lowers the time to load a new molecule and will optimize the duration of the measurements.

## 4. Conclusions

Single-molecule recycling (SMR) enables prolonged observation of a single molecule by keeping the molecule in dark for most of the period to avoid non-reversible photobleach and reversible dark state. This paper proposed SMR in a capillary microchannel to reduce the experimental difficulties of SMR in a nanochannel such as non-specific sticking of molecules to the surface, low likelihood of locating a molecule in the nanochannel, and high requirements for the preparation for the walls of the nanochannel device [26]. By translating the capillary with a computer-controlled piezo stage and fitting the histograms of time gaps between the peaks of photon bursts and the starts of piezo translations, we can estimate the diffusivity from the displacements of a molecule with respect to the capillary and generate result of the same magnitude as the actual value. The diffusivity of a molecule depends on its properties such as size and shape, and the viscosity of the suspending medium. SMR offers a method for measuring the diffusivity in one dimension, which differs from that in bulk solution [22,29], and provides an unique insight into the interaction between molecules, i.e., the slow-down in the diffusivity of fluorescent ligands when they become bound to larger biomolecules [21]. We use SMR in a capillary microchannel as a substitute to SMR in a nanochannel on occasions where only the magnitude of diffusivity is concerned, which would be possible in pharmaceutical drug discovery research.

## Figures and Tables

**Figure 1 micromachines-12-00800-f001:**
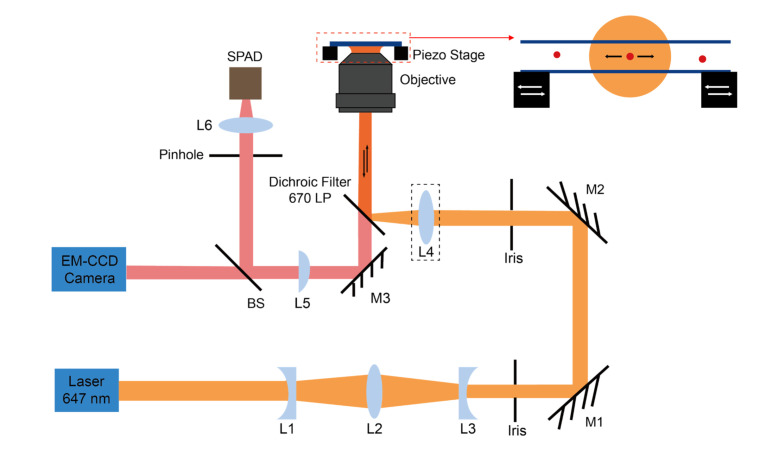
Schematic of the confocal microscope system: The laser beam is expanded by L1-3 (L1, 150 mm focal length, plano-concave lens; L2, 100 mm focal length, bi-convex lens; L3, 75 mm focal length, plano-concave lens) to fit the entrance pupil of the objective and passes through an Iris and is then raised by mirrors M1-2 to the level of the dichroic filter. L4 (150 mm focal length, biconvex lens) is a removable Kohler lens placed at the back focal point of the objective to achieve Kohler illumination. The laser beam is reflected by the dichroic filter to the objective, the collected fluorescence passes through the dichroic filter and is focused with L5 (250 mm focal length, plano-convex lens). The beam is then split by beam sampler (BS) to an EM-CCD camera and through a pinhole to focus on an sing-photon avalanche diode (SPAD). The inset highlights the piezo stage and the microchannel device, where the molecule diffuses through the focal volume and the microchannel is moved back and force by the piezo stage. The capillary is translated to one end of the recycle and is moved back after a fix reversal delay reduced by the translation time. Therefore, the molecule is moved through the focal volume after each reversal delay. The combined movement of the molecule resembles that of SMR in a nanochannel, where the molecule is driven by the electrophoresis.

**Figure 2 micromachines-12-00800-f002:**
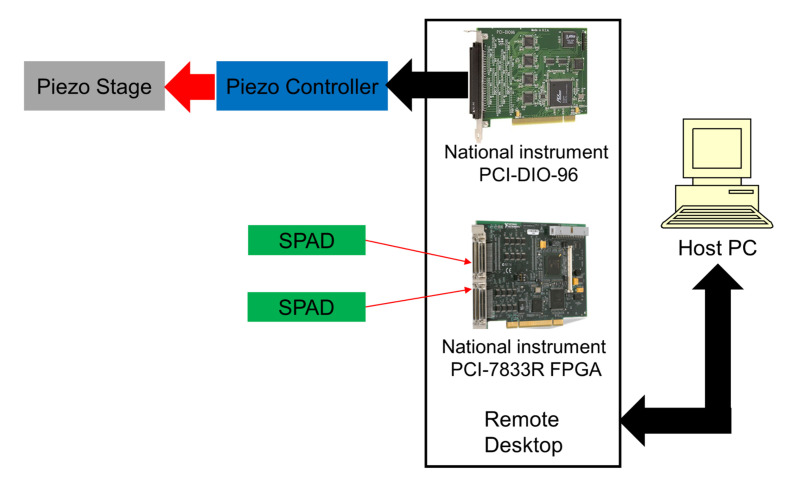
Description of the real-time control system.

**Figure 3 micromachines-12-00800-f003:**
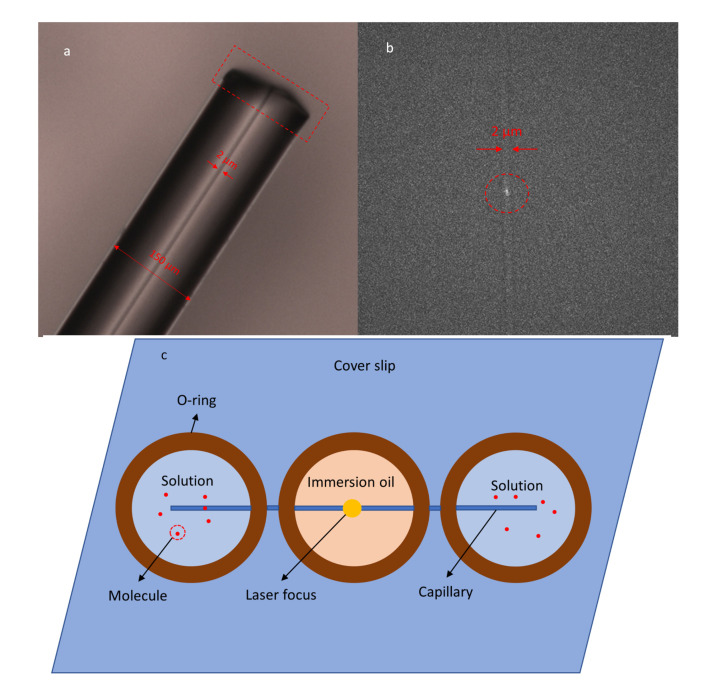
(**a**) The size of the capillary tubing. The red box highlights the chopped end of the capillary. (**b**) The laser focus is adjusted to the inner diameter of the capillary. The red circle highlights the laser focus inside the tube. The image is taken by the EM-CCD camera. (**c**) Schematic of the capillary microchannel device. The o-rings on the sides are reservoirs of solution, and the o-ring in the middle is the observation window for the objective. This o-ring is filled with immersion oil to reduce refraction and reflection from the outer wall of the capillary.

**Figure 4 micromachines-12-00800-f004:**
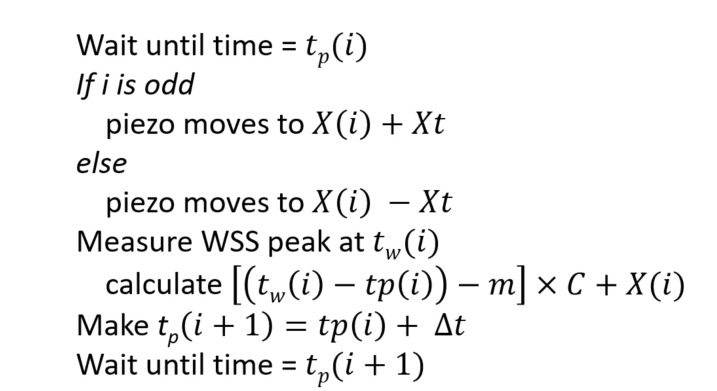
The algorithm applied in the SMR with the piezo stage.

**Figure 5 micromachines-12-00800-f005:**
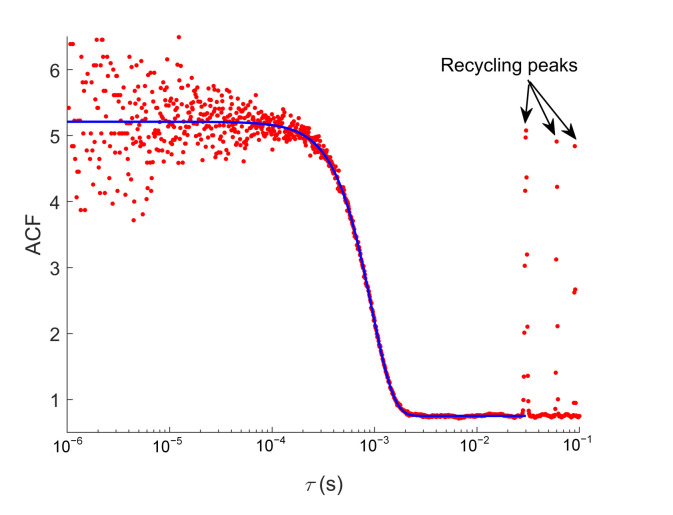
ACF of immobilized fluospheres in SMR. The laser beam is adjusted to fit the inner diameter of the capillary microchannel. The peaks on the right of the slope represent the intervals between passages in SMR, which are 30 ms, 60 ms, and 90 ms, respectively. The curve of ACF is fitted by Equation (Equation 7).

**Figure 6 micromachines-12-00800-f006:**
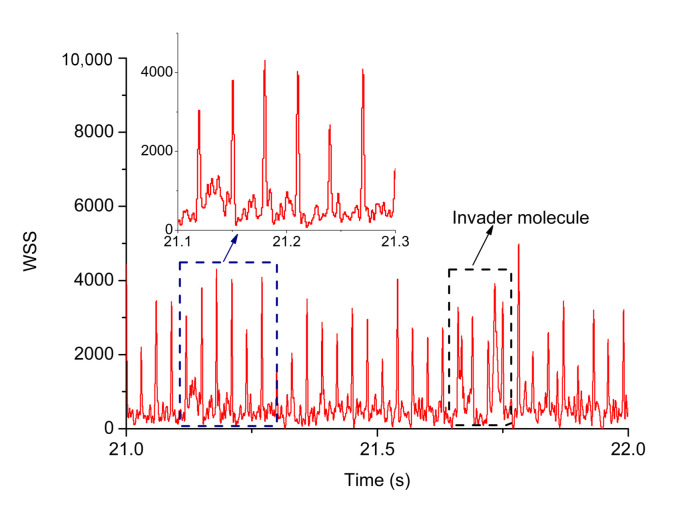
WSS of a fluosphere that diffuses through the capillary and is recycled by a piezo stage. The inset illustrates the intervals between fluorescence peaks, which center around reversal delay, 30ms and are differed from error induced by diffusion. The black box around 22.7 s describes a second molecule diffuses to the region of recycling and disrupts the detection of passing times. The algorithm tolerates the invader for several cycles and eventually the invader diffuses out of the region of recycling.

**Figure 7 micromachines-12-00800-f007:**
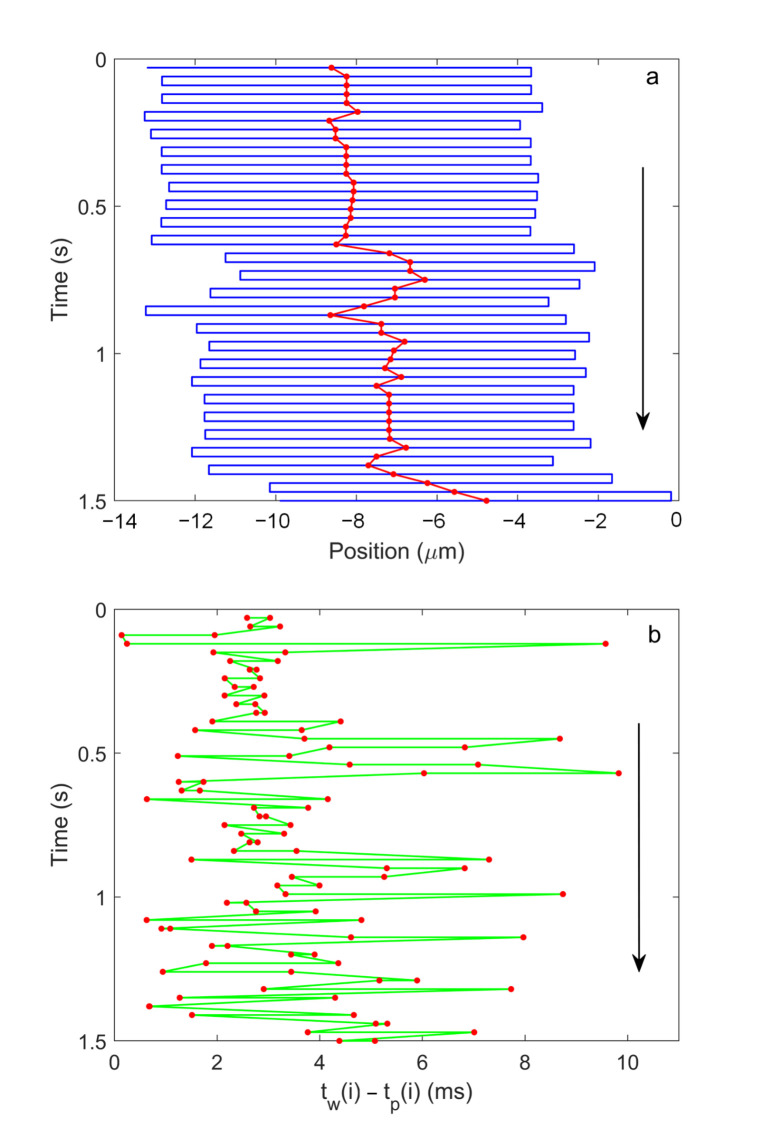
(**a**) Blue solid line is the trajectory of the piezo stage, the red dot is then center of the recycling. (**b**) The red dot is $t_{w}\left(i\right)-t_{p}\left(i\right)$ in each recycle.

**Figure 8 micromachines-12-00800-f008:**
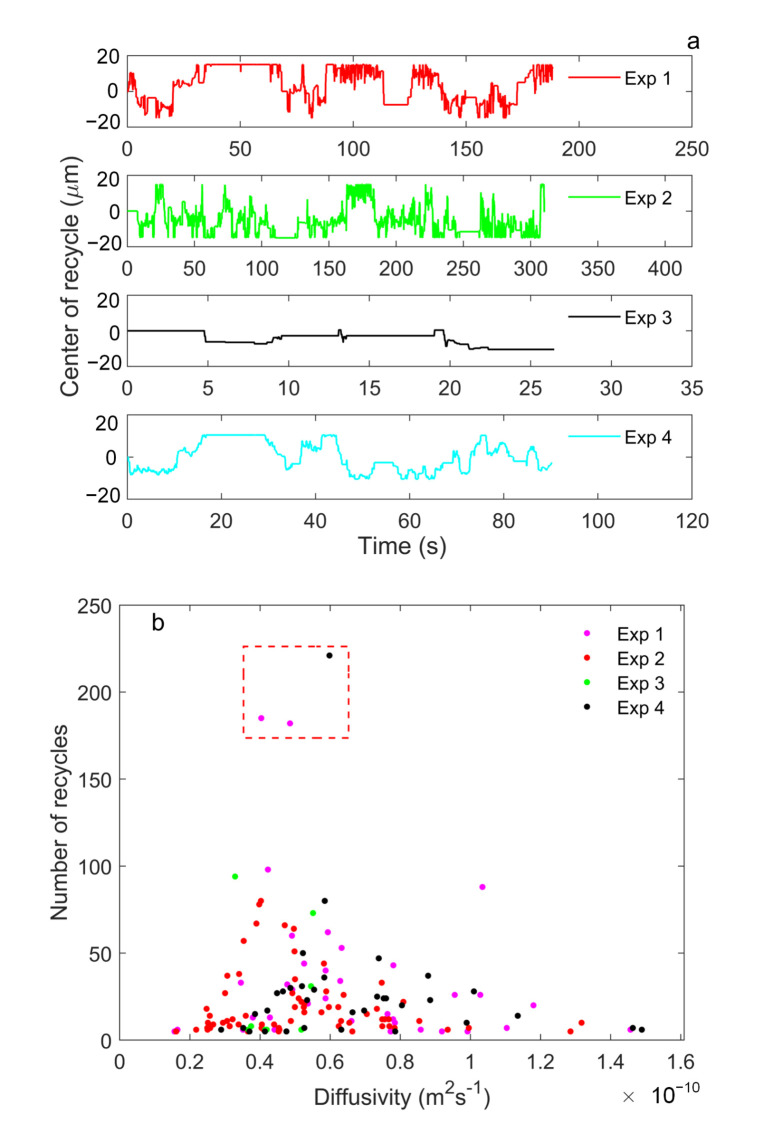
Results from SMR in the capillary microchannel with the piezo stage. (**a**) The variations of centers of recycles in 4 experiments. In each experiment, multiple molecules come into recycling and the periods without fluctuations represent the piezo stage stops and waits for the next molecule to be recycled. The maximum number of recycles is above 200, which is equivalent to longer than 6 s. (**b**) The estimated diffusivities of individual FLuospheres, which forms a Gaussian distribution centered around $5\times 10^{-11}\;m^{2}s^{-1}$. The red box highlights the molecules being recycled for about 200 times.

**Figure 9 micromachines-12-00800-f009:**
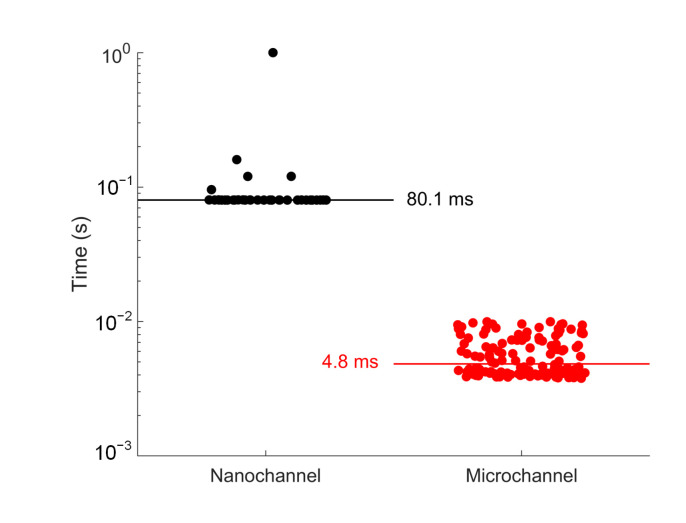
The time to locate a new molecule during the SMR experiment. The horizontal line in each scatter plot represents the median of the time.

## Data Availability

The simulation and data analysis code used in this paper can be found at “https://github.com/RocketSci2014”.

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
