# Peer review of "Diffusivity Measurement by Single-Molecule Recycling in a Capillary Microchannel"

_micromachines, 2021, doi:10.3390/mi12070800_

Round 1

Reviewer 1 Report

Dear authors,

Please update the manuscript with the comments suggested below to improve the quality of the manuscript.

  1. The abstract of a scientific research paper should be precisely mentioning the specific research question that is answered, experimental conditions, operational parameters, results, and conclusions. Please explain and provide the specifications of the scientific questions answered in the study performed.
  2. In the abstract, please include the details of the limitations of the current/previous technologies/studies and specify the scientific advancements made in the diffusivity measurement by using single-molecule recycling in a capillary microchannel, to overcome those limitations.
  3. The abstract needs to be incorporated with the gist of the complete work in the manuscript. In the current version of the abstract, only the summary of the work is mentioned in a very broad view. Instead, please include the specific details of the research study presented in the manuscript.
  4. In the current version of the abstract, the scientific conclusions made from the study performed on the diffusivity measurement by single-molecule recycling are not specific. Kindly detail the conclusions that are driven from the study performed.
  5. In the introduction, please include more details of the peer studies performed on the ‘single-molecule recycling in the capillary microchannel’, to guide the reader to understand the importance of the research study performed. Also, include corresponding references in the text when mentioning the details.
  6. In the introduction, please include the knowledge gaps existing between the current research work and prior studies performed in the field. Very importantly, please specify the need for the current work presented in the manuscript.
  7. In the last paragraph of the introduction, kindly include the details of the broader impacts on the study made and the results achieved. It is very important to provide the future scope of the research performed to make a strong impact on the readers on the research performed.
  8. Please update figures from 5 to 9 with high-quality images. The figures in the current version of the manuscript are very low in quality and difficult to review.
  9. In section 3.3, kindly incorporate the logical reasoning and scientific conclusions made from the plots in figures 7,8 & 9, also please use the ongoing research results of peers with appropriate references to support your arguments and statements.
  10. Please revise the manuscript with English grammar. There are few places that the manuscript needs to be improved with respect to English writing.

Reviewer 2 Report

  1. The mathematic model and closed-form equations do not make significant needs to the part shown later.

  2. Figure 2 does not look good-quality representation and relate to the necessary.

  3. Figure 4 copies the algorithm.

  4. Fig.7 and 9 look like improvement is needed.

  5. The quality of this manuscript might need much improvement. I hardly support this manuscript for publication.

Reviewer 3 Report

The authors developed a novel piezo-based system coupled with capillaries to dected the difusivity of single molecules with much more accurate time controlling. The background was presented well, the data are solid, and the calculations are reasonable. This is significant work in the field because of the shorter times. I read through the manuscript and did not find any necessary revision needs. 

Author Response

Thank you so much for the review!

Reviewer 4 Report

In the article entitled: " Diffusivity measurement by single molecule recycling in a capillary microchannel “ authors Bo Wang and Lloyd M. Davis presented method and results of measurements of Brownian motion diffusion coefficient in microchannels. From my point of view, the adopted methodology of performing experiments and processing their results do not raise any objections.

My guess is that the prepared research stand is to be used to conduct experiments using various forces (including electrostatic forces) forcing the movement of molecules. Probably this is the purpose of the electrodes shown in Figure 3. The lack of any mention of how they are connected may raise some doubts as to the induction of voltage in them. After all, in the research apparatus there are variable electrostatic fields that control the piezoelement. The charging and discharging currents of its capacity are relatively large and by induction they can counteract or support the movement of molecules. I believe that there should be some clarification in the paper regarding this issue.

The system for controlling the xyz position of the measuring system (capillary), mentioned in line 177, may cause slightly smaller, but justified doubts. The main point is whether or not it may or may not be a source of measurement disturbances.

While reading the text of the paper, I encountered a few shortcomings that should be corrected before its publication:

line 13

Due to the less advanced readers, the abbreviation FPGA (Field Programmable Gate Array) should be deciphered. This should not necessarily be done in the abstract.

line 46

I suggest replacing “of the nanoscale inner diameter” with “of the tube with nanoscale inner diameter”.

line 47

I suggest replacing “contracted” with “lowered”.

lines 51-52

This part of the sentence is unclear: "Because the inner diameter of the channel is significantly larger than a nanochannel and the length of the channel is in the scale of millimeters". This needs to be corrected.

lines 50-65

The introduction should include a literature review and the methods described therein. Therefore, the information on the research methods described in the paper should be transferred to the next chapter.

line 87

It would be advisable to provide the literature source of the Fokker-Planck equation.

line 88

The abbreviation pdf (probability density function) should be deciphered in the text.

line 90

The function  should be described in the text.

line 92

It is probably better to use the word "displacement" instead of "moving".

line 101

I suggest replacing “minus” with “reduced by”.

Formulae (3)

I suggest replacing “sqrt” with square root sign.

lines 111 and 242

I suggest replacing “,” with “:”.

Formulae (5)

Please remove dot following relation.

line 116

I suggest replacing “described by” with “shown in”.

line 153

I suggest replacing “hose” with “host”.

line 165

I suggest replacing “translated” with “moved”.

line 257

I suggest replacing “ms” with “ms-1”.

Formulae (12)

In the dependent variable on the left side of equation (12), the index i should be removed.

Caption under Figure 9

There is a mistake in the caption under Figure 9. The power value in the given centered Gaussian distribution should be -11.

line 317

I suggest replacing “the difficulties” with “the experimental difficulties”.

line 319

I suggest replacing “high requirements to manufacture the nanochannel device” with “high requirements for the preparation of the walls of the nanochannel device”.

Reviewer 5 Report

This paper concerns measuring diffusivity of single molecules recycling in a microcapillary. A significant portion of the paper is similar to the authors’ paper published in 2017. While there is merit in the proposed design, novel results are too few and my recommendation is that the paper at this stage is not suited to publication in Micromachines. The comments are given below.

  1. Introduction: Strongly suggest the introduction is improved. It is not immediately clear what is new in this paper. For example, in line79 it is mentioned that this paper is similar to reference 25. The authors should explain exactly how it is similar and how it is different.
  2. Section 2.6: Multiple typos should be revised.
  3. Line 106-116 : Fig. 1b is mentioned in the text before Fig. 1a.
  4. 1a: Maybe provide more detail about the rest of the setup instead of just the confocal microscope.
  5. 1b: The explanation is not clear for SMR mechanism. Please explain more clearly.
  6. 3a, Fig. 3b and Fig 5: These figures are exact copies of those found in “Improved timing and diffusivity measurement in single-molecule recycling in a nano-channel” published by the same authors in 2017. Use reference whenever needed or use different figures.
  7. Line 316: authors mention the advantages of a microchannel versus a nanochannel as one of the main contributions of the paper; therefore, their paper could greatly benefit from adding data that shows how the results (as well as the likelihood of locating the molecule) would be affected based on channel dimensions.

Round 2

Reviewer 1 Report

Dear authors,

Thank you for updating the manuscript with recommended changes.

Reviewer 2 Report

No more suggestion.

Reviewer 5 Report

The revised manuscript has been improved for further publication.